# Interferon Gamma-Inducible NAMPT in Melanoma Cells Serves as a Mechanism of Resistance to Enhance Tumor Growth

**DOI:** 10.3390/cancers15051411

**Published:** 2023-02-23

**Authors:** Cindy Barba, H. Atakan Ekiz, William Weihao Tang, Arevik Ghazaryan, Mason Hansen, Soh-Hyun Lee, Warren Peter Voth, Ryan Michael O’Connell

**Affiliations:** 1Division of Microbiology and Immunology, Department of Pathology, University of Utah, Salt Lake City, UT 84112, USA; 2Izmir Institute of Technology, Molecular Biology and Genetics Department, Gulbahce, Izmir 35430, Turkey

**Keywords:** melanoma, NAMPT, interferon gamma

## Abstract

**Simple Summary:**

The tumor microenvironment is complex, with interacting immune and tumor cells. Immune cells release inflammatory cytokines, including interferons (IFNs), that drive tumor clearance. However, evidence suggests that tumor cells can also utilize IFNs to enhance growth and survival in certain cases. We demonstrate that interferon gamma (IFNγ) mediates the metabolic reprogramming of melanoma cells by inducing the essential NAD+ salvage pathway enzyme nicotinamide phosphoribosyltransferase (NAMPT) gene through STAT1 binding to the NAMPT locus. NAMPT is constitutively expressed in cells during normal homeostasis. However, melanoma cells have higher energetic demands and increased NAMPT. We show that IFNγ signaling upregulates NAMPT in melanoma cells, increasing cell proliferation and survival. Further, STAT1-inducible *Nampt* promotes melanoma growth in mice. We provide evidence that melanoma cells directly respond to IFNγ-activated STAT1 by increasing Nampt, which improves their fitness during tumor immunity. Elucidating mechanisms that regulate NAMPT expression can lead to enhanced therapeutic approaches with immunotherapies that utilize IFN signaling to improve patient outcomes.

**Abstract:**

(1) Background: Immune cells infiltrate the tumor microenvironment and secrete inflammatory cytokines, including interferons (IFNs), to drive antitumor responses and promote tumor clearance. However, recent evidence suggests that sometimes, tumor cells can also harness IFNs to enhance growth and survival. The essential NAD+ salvage pathway enzyme nicotinamide phosphoribosyltransferase (NAMPT) gene is constitutively expressed in cells during normal homeostasis. However, melanoma cells have higher energetic demands and elevated NAMPT expression. We hypothesized that interferon gamma (IFNγ) regulates NAMPT in tumor cells as a mechanism of resistance that impedes the normal anti-tumorigenic effects of IFNγ. (2) Methods: Utilizing a variety of melanoma cells, mouse models, Crispr-Cas9, and molecular biology techniques, we explored the importance of IFNγ-inducible NAMPT during melanoma growth. (3) Results: We demonstrated that IFNγ mediates the metabolic reprogramming of melanoma cells by inducing Nampt through a *Stat1* binding site in the *Nampt* gene, increasing cell proliferation and survival. Further, IFN/STAT1-inducible Nampt promotes melanoma in vivo. (4) Conclusions: We provided evidence that melanoma cells directly respond to IFNγ by increasing NAMPT levels, improving their fitness and growth in vivo (control n = 36, SBS KO n = 46). This discovery unveils a possible therapeutic target that may improve the efficacy of immunotherapies involving IFN responses in the clinic.

## 1. Introduction

Melanoma is the most common form of cancer in the U.S., increasing in prevalence over the last decade [1]. This form of skin cancer is characterized by the uncontrolled growth of melanocytes and is treated with surgical resection, chemotherapy, or immunotherapy, depending on the tumor stage [2]. Melanomas frequently display resistance to such therapeutic approaches; therefore, new avenues for treating melanoma are urgently needed [3]. 

In melanoma, immune cells infiltrate the tumor microenvironment (TME) and secrete inflammatory cytokines, such as interferons (IFNs) [4]. Type I and II IFNs reduce tumor growth and have significant anti-proliferative effects [4,5,6]. Type II IFNs, such as IFNγ, are hallmark cytokines produced by tumor-infiltrating lymphocytes (TILs), such as NK and T cells, and their activation leads to tumor clearance through multiple mechanisms [7]. Recently, there has been a new appreciation for mechanisms by which tumor cells evade immune responses. Although counterintuitive, IFNs also have pro-tumorigenic roles in cancer [5,7,8] and enhance tumor growth by promoting angiogenesis [9], upregulating co-inhibitory molecules involved in immune checkpoints, including the programmed death ligand 1 protein 1 (PD-L1) [10], and preventing apoptosis [11]. These contrasting findings require further characterization of IFNγ signaling to optimize therapeutic windows.

Here, we examined a key nexus of IFNγ-mediated tumor cell fitness, metabolic reprogramming by upregulating the essential nicotinamide adenine dinucleotide (NAD+) salvage pathway enzyme, nicotinamide phosphoribosyltransferase (NAMPT) [12]. All cells utilize NAMPT as the rate-limiting enzyme in NAD+ production, and it is important in biological signaling and ATP production [12]. In addition, tumor cells divide rapidly, leading to increased metabolic demands and an increased dependence on NAD+, a cofactor central to metabolism [13,14]. Melanoma cells have increased NAMPT expression [15], and NAMPT inhibition significantly reduces melanoma cell proliferation [12,15,16]. Concordantly, NAMPT inhibitors, such as FK866, have been tested as a novel cancer therapy and showed reduced tumor burden in preclinical studies, leading to several clinical trials (NCT00432107, NCT00431912) [17]. NAMPT is clearly important for cell homeostasis, and we have identified a novel IFNγ-induced mechanism by which NAMPT enhances tumor burden. 

Our data elucidates a mechanism by which IFNγ upregulates NAMPT in both human and mouse melanoma cells by inducing the signal transducer and activator of transcription 1 (STAT1) transactivation via a conserved enhancer that we recently identified in tumor-associated macrophages [18]. Further, IFN-inducible NAMPT contributes to melanoma cell growth in vitro and in vivo. Despite the well-established anti-proliferative effects of IFNs on tumor cells, our findings reveal novel facets of IFNγ signaling in tumor cells that may contribute to the pathogenic effects mediated by antitumor immunity, as seen in recent studies [5]. Together, these data suggest an essential role for IFN-inducible NAMPT in melanoma progression. By identifying mechanisms by which IFNs and IFN-based therapies promote tumor growth, the efficacy of such treatments in the clinic will be enhanced. 

## 2. Methods 

### 2.1. Cell Cultures and Reagents

Mouse B16-F10 melanoma cell lines were grown in DMEM supplemented with 10% FBS, 1% penicillin-streptomycin, and 1% L-glutamine at 37 °C in a humidified atmosphere containing 5% CO_2_. Mouse YUMM (1.1, 3.2 and 5.2) melanoma cell lines were cultured in DMEM/F12 media containing 10% FBS and supplemented with 1% pen-strep and 1% non-essential amino acids. Lipofectamine RNAi MAX and Opti-MEM medium were purchased from Invitrogen. FK866 was obtained from Sigma, IFNγ was obtained from Ebioscience, and NMN was obtained from Sigma-Aldrich. NAD+ was measured using the NAD+/NADH quantitation kit from Sigma according to the manufacturer’s instructions.

### 2.2. In Vitro Activation, NAMPT Inhibition, and Agonist Treatment

Tumor (B16-F10, YUMM 1.1, YUMM 3.2, YUMM 5.2, A375, A2058) mouse and human cell lines were cultured in their respective media, as mentioned under Cell cultures and reagents. For the IFNγ stimulation/NAMPT inhibition experiments, B16-F10 cells were stimulated with either mouse or human IFNγ (50 ng/mL mouse IFNγ, Ebioscience Cat # 14-8311-63; 50 ng/mL human IFNγ, Ebioscience Cat # 14-8319-80) and FK866 (50 nM, R&D Systems Cat # 4808/10) for 6 h for RNA isolation or 24 h for protein isolation. B16-F10 cells were also treated with 10 ng/mL of IFNβ in respective experiments. In experiments with IFNγ stimulation only, cells were stimulated with 50 ng/mL IFNγ, unless otherwise indicated. Where indicated, 100 µM NMN was added to cell culture at the beginning of each experiment to rescue the effects of FK866 on the depletion of NAD+. C1498 and MC38 cells were treated with IFNγ (50 ng/mL mouse IFNγ, Ebioscience Cat # 14-8311-63) for 6 h for RNA isolation qPCR analysis.

### 2.3. RNA Isolation from Cultured Cells and RT-qPCR 

Cells were lysed with Qiazol reagent prior to the isolation of total RNA with a miRNeasy RNA isolation kit (Qiagen, Germantown, MD, USA). An amount of 200–400 ng of RNA was utilized in a cDNA synthesis reaction with qScript kit reagents (Quanta Biosciences, Beverly, MA, USA) and was diluted to a final volume of 200 μL. Quantitative PCR (qPCR) was performed in a QuantStudio 6 (Thermo, Madison, WI, USA), utilizing Promega GoTaq qPCR master mix reagents, or the equivalent, and qPCR primers listed in Appendix A. RT-qPCR data presented were representative of at least 2 independent experiments with at least 2 biological replicates per condition and 2 to 3 technical replicate PCR reactions per sample.

### 2.4. CRISPR/Cas9-Mediated Gene Deletion of the STAT1 Binding Site in Tumor Cell Lines

CRISPR/Cas9 lentivector infections were performed by utilizing Trans-IT 293 (Mirus, Madison, WI, USA) to transfect 293T cells with the packaging plasmids pVSVg and psPAX2, along with a modified CRISPR plasmid based on lentiCRISPRv2 (Addgene plasmid # 52961), as described [19,20]. The deletion of the STAT1 binding site was made by targeting sequences immediately flanking the SBS region expressed from a single lentiCRISPRv2 vector containing a dual promoter system, designed at the University of Utah Mutation Generation and Detection Core, and consisting of the U6 promoter for one gRNA and the H1 promoter for the second. Clonal cell populations were obtained by plating single cells on a 96-well plate, and the expression of the NAMPT was quantified by IFNγ stimulation following RNA isolation and RT-qPCR to verify the knockout of the STAT1 binding site. Controls for each of these cell lines were generated in a similar manner as described above, with a CRISPR/Cas9 lentiCRISPRv2 vector plasmid that contained non-targeting sgRNA sequences. 

### 2.5. Western Blots

Cell pellets were lysed in RIPA buffer containing 150 mM Tris (pH7.4), 150 mM NaCl, 1% NP40, 1 mM EDTA, and 1% SDS. The purified protein was separated via SDS-PAGE and transferred to a 0.45 µM nitrocellulose membrane. All experimental samples and controls used for one comparative analysis were run on the same blot/gel. Antibody staining of NAMPT (Bethyl Laboratories) and ACTB (Santa Cruz sc-47778) was performed. Primary antibody binding was detected with IRDye-700- or IRDye-800-conjugated secondary antibodies (LI-COR, Lincoln, NE, USA) using a LI-COR Odyssey Infrared Flatbed Scanner. Antibodies used are detailed in Appendix A. 

### 2.6. Mice

All mice were on a C57BL/6J genetic background obtained from The Jackson Laboratory (Bar Harbor, ME, USA), cataloged as C57BL/6J, Stock No: 000664. All experimental procedures using mice were performed with the approval of the Institutional Animal Care and Use Committee of the University of Utah.

### 2.7. B16-F10 Melanoma Experiments 

Mice were shaved, at most, 24 h prior to any injection. Mice were then injected subcutaneously in the rear flank with 5 × 10^5^ B16-F10 melanoma cells on day 0. Tumor size measurements were taken on the indicated days, and tumors were weighed upon sacrifice at day 14. 

### 2.8. SBS Mutation Sequencing 

The DNA template for amplicon sequencing of SBS mutationswas obtained from the control and SBS KO B16-F10 cells that were cultured as mentioned above. Cells were treated with IFNγ (50 ng/mL mouse IFNγ, Ebioscience Cat # 14-8311-63) and/or FK866 (50 ng/mL, R&D Systems Cat # 4808/10) for 6 h. PCR using primers containing adaptor sequences according to manufacturer’s instructions and targeting the SBS region of NAMPT was performed. For each clone, 4 separated aliquots were individually amplified to reduce sampling error during PCR. After the confirmation of the correct amplicon by gel electrophoresis, separate aliquots were pooled together and then purified using PCR purification (Qiagen Cat # 28106). Individual SBS KO clones and pooled control cell clones were then submitted for amplicon sequencing (Amplicon-EZ sequencing, Genewiz Inc., South Plainfield, NJ, USA).

### 2.9. Flow Cytometry

After the mechanical disruption of tumor tissue, tumor cells were placed on an orbital shaker in Accumax (Innovative CellTechnologies) and incubated for 30 min at room temperature, followed by red blood cell lysis in ammonium–chloride–potassium buffer (Biolegend) and filtration through a 0.45 micron filter (18). Flow cytometric analyses were performed on single-cell suspensions isolated from tumors. Prior to staining with fluorophore-conjugated antibodies (all from Biolegend or Thermo/eBioscience), Fc receptors were blocked using αCD16/32 antibody (1:200, Biolegend) to reduce nonspecific staining. Staining was performed on ice for 15–30 min in PBS supplemented with 2% FBS and 0.1% sodium azide. Cells were stained with a combination of the following fluorophore-conjugated antibodies in Hanks’ balanced salt solution supplemented with 10% BSA, pyruvate, EDTA, and HEPES. Cultured cells were treated with various reagents as described and were collected off the plate via 5 mM EDTA in PBS in a non-enzymatic manner. For viability assays, Ghost Dye Red 780 (Tonbo) stain was applied. The cells were then washed, and flow cytometry was performed using a BD LSRFortessa II (BD Biosciences). For death assay, 7AAD-PerCP Cy5.5 (2 µL/test) and Annexin V-PeCy7 (2 µL/test) were stained in annexin binding buffer for 30 min. The cells were then washed, and flow cytometry was performed using a BD LSRFortessa II (BD Biosciences). Data were subsequently analyzed using FlowJo software, utilizing the gating strategy shown in Appendix A (Tree Star). Antibodies used are detailed in Appendix A.

### 2.10. Incucyte Live Image Analysis

Tumor cells were cultured at a seeding density of 5000 cells per well in a 96-well plate for 24 h to allow cell adherence. Media was then changed to add appropriate drug conditions: IFNγ (50 ng/mL mouse IFNγ, Ebioscience Cat # 14-8311-63; 50 ng/mL human IFNγ, Ebioscience Cat # 14-8319-80), FK866 (50 ng/mL, R&D Systems Cat # 4808/10), and NMN (100 µM, Sigma Aldrich, Inc. Product # N3501). Cells were then placed in the incubator containing the Incucyte at 37 °C in a humidified atmosphere containing 5% CO_2_ for up to 140 h. Phase confluence was determined using the IncuCyte Zoom software. Four images of each well were taken every 2 h and normalized to phase confluence plotted over time.

### 2.11. Statistical Analysis

T-tests were performed when comparing differences between two experimental groups in the in vitro experiments. Tumor growth rates were analyzed using 2-way ANOVA comparison. Unless otherwise noted, the following symbols were utilized in all figures to denote significant *p*-values: * *p* ≤ 0.05, ** *p* ≤ 0.01, *** *p* ≤ 0.001, **** *p* ≤ 0.0001, and ns for not significant. 

## 3. Results

### 3.1. IFNs Induce NAMPT in Melanoma Cells

Our previous work identified that IFNγ upregulates NAMPT in tumor-associated macrophages and is important for their anti-tumorigenic function [18]. Additionally, melanomas have higher levels of NAMPT [15]; therefore, we investigated whether IFNs induce NAMPT in melanoma cells. We treated B16-F10 mouse melanoma cells with IFNγ for 6 or 24 h, which significantly increased NAMPT mRNA expression (Figure 1A). In addition, 24 h of IFNγ treatment induced higher levels of the NAMPT protein in B16-F10 cells (Figure 1B), with Western blot quantification showing significant increases in protein levels (Figure 1C). To examine the IFN–NAMPT axis in other clinically relevant models, we used Yale University Mouse Melanoma (YUMM) cells expressing mutations commonly observed in melanoma patients [21]. We treated YUMM 1.1, 3.2, and 5.2 mouse melanoma cell lines (Figure 1D) and A375 and A2058 human melanoma cells (Figure 1E) with IFNγ. In all cases, IFNγ significantly induced NAMPT expression in mouse and human melanoma cells. Next, we determined if NAMPT was induced in other cancer types in response to IFNγ. We treated C1498, a leukemia cell line, and MC38, a colon cancer cell line, with IFNγ. NAMPT was upregulated in C1498 but not in MC38 cells, suggesting tumor type dependency in the IFNγ-induction of NAMPT (Figure 1F). In addition, we tested whether IFNβ induced NAMPT in melanoma cells, since it has been shown to be within the TME and can increase pro-tumor factors [10,22]. In all cases of cells treated with IFNγ, CXCL10 was measured to verify IFNγ induction (Appendix A). We found that B16-F10 cells treated with IFNβ had a significant upregulation of NAMPT (Figure 1G), suggesting that a variety of IFNs can induce NAMPT. Altogether, we showed strong evidence that IFNs induce NAMPT in mouse and human melanoma cells, suggesting that the IFN-mediated metabolic changes are, in part, mediated by differentially regulated NAMPT. 

### 3.2. IFNγ-Inducible NAMPT Plays an Important Role in Melanoma Cell Growth

Previous literature implicates NAMPT in melanoma proliferation [16]. After observing the IFN induction of NAMPT expression in melanoma cells, we investigated whether IFNγ-inducible NAMPT promoted proliferation in vitro. We treated B16-F10 cells with FK866, a chemical inhibitor of NAMPT enzymatic function, and nicotinamide mononucleotide (NMN), the direct byproduct of NAMPT, and measured the proliferation of cells with each individual treatment or with a combination of the three using Incucyte live imaging analysis for 6 days. Melanoma cells treated with either IFNγ or FK866 had a reduction in growth that was further exacerbated upon combination treatment with IFNγ and FK866 (Figure 2A). To rescue the effect of NAMPT, FK866, we treated cells with FK866 and NMN, restoring growth and validating the function of NAMPT (Figure 2A). These results were consistent with previous works and further suggest that IFNγ-inducible NAMPT is important for melanoma proliferation [16]. Melanoma cells also exhibited a significant decrease in cell confluency when treated with IFNγ and FK866, and we sought to identify if it was due to cell death or an inhibition of proliferation (Figure 2B). We treated B16-F10 cells with IFNγ, FK866, or a combination of the two and analyzed our data using flow cytometry (Figure 2C). To quantify proliferation, we used Cell Trace Violet (CTV), and to quantify cell death, we used 7AAD and Annexin V staining. The CTV fluorescent signal was stratified into high and low subsets using flow cytometry. IFNγ/FK866 combination-treated melanoma cells had the highest proportion of CTV-high (non-proliferating) cells and the lowest proportion of CTV-low (proliferating) cells, suggesting that IFNγ with FK866 decreased growth more significantly than either treatment alone (Figure 2D). In addition, the mean fluorescence intensity (MFI) of CTV was highest in the IFNγ/FK866 combination-treated cells, further suggesting that the combination treatment had the greatest reduction in melanoma cell proliferation (Figure 2E). Altogether, we conclude that IFNγ-inducible NAMPT is an important player in melanoma cell growth. After observing that IFNγ/FK866 combination-treated cells grew significantly slower, we wanted to investigate if cell death played a role in the growth reduction. To do so, we treated B16-F10 melanoma cells with IFNγ, FK866, or a combination of the two and analyzed our data using flow cytometry to quantify live and dead cell populations. IFNγ/FK866 combination-treated cells also exhibited the lowest proportion of live cells and the highest proportion of dead cells using a live/dead stain (Figure 2F). Additionally, we used this same technique with another stain combination to further characterize the type of cell death that was induced by IFNγ/FK866 treatment. Cells treated with the combination of IFNγ and FK866 had the highest proportion of early apoptotic and late apoptotic cells and the lowest proportion of necrotic cells (Figure 2G). Altogether, these data indicate that the inhibition of IFNγ-induced NAMPT leads to cell death in melanoma.

### 3.3. Mutations in NAMPT STAT1 Binding Sites Lead to Significantly Lower Levels of IFNγ-Dependent NAMPT Induction

IFNγ and FK866 combination treatment significantly decreased melanoma growth, but we wanted to assess the contribution of IFNγ-inducible NAMPT in melanoma growth, specifically. We previously identified a novel conserved STAT1 binding element in the *NAMPT* locus that was sufficient for the IFNγ-mediated induction of *NAMPT* in macrophages [18]. Our previous publication outlined the predicted Stat1 consensus binding sites in the first intron of the NAMPT gene (Figure 3A). Previously, we validated that deletion of the first two binding sites led to decreased IFNγ-inducible NAMPT while maintaining the baseline expression of NAMPT [18]. Therefore, we utilized this information and deleted the STAT1 binding sites in B16-F10 cells (SBS KO) with a lentiviral CRISPR/Cas9 knockout vector expressing two sgRNAs targeting flanking sequences of the SBS element. We compared cells with our CRISPR construct to control cells expressing non-targeting sgRNAs (Figure 3A). A spectrum of CRISPR-Cas9-driven mutations were seen in the analyzed clones, confirming the successful deletion of the Stat1 binding site of *NAMPT.* By analyzing multiple lines cloned from single cells, we confirmed that, upon IFNγ stimulation, SBS KO cells significantly reduced *NAMPT* induction (Figure 3B).

### 3.4. IFNγ-Dependent NAMPT Induction Plays a Tumor Cell-Intrinsic Role in Promoting Melanoma Growth In Vivo

Lastly, we sought to investigate whether IFN/Stat-inducible NAMPT regulates melanoma growth in vivo. To do so, we injected C57BL/6J WT mice with either control or SBS KO B16-F10 tumor cells subcutaneously and monitored tumor size and composition. Control and SBS KO tumors were measured every two days and weighed at the end of the experiment. Altogether, SBS KO cells grew significantly smaller tumors, compared to wild-type B16-F10 cells, over time by volume and mass (Figure 3C,D). In addition, we immunologically charactered the TME of wild-type and SBS-KO tumors using flow cytometry. We quantified the percent of overall immune cells (CD45+), T cells (CD3+), helper T cells (CD4+), cytotoxic T cells (CD8+), B cells (B220+), and macrophages (CD11b+ F4/80+) within tumors from control and SBS KO B16-F10-bearing mice and found no overall differences in immune composition (Figure 3E,F, Appendix A). We also assessed programmed cell death protein 1 (PD1) expression on T cells and programmed cell death ligand protein 1 (PD-L1) within the tumor to rule out tumor growth differences attributed to the immune function within the TME. We found no differences in T cell PD1 expression between the control and SBS KO tumors (Figure 3G, Appendix A). Additionally, we found no differences in non-immune cell PD-L1 (CD45- PD-L1+) expression between the control and SBS KO tumors and modest but significant increases in the mean fluorescence intensity (MFI) of non-immune cell PD-L1 (Figure 3H,I, Appendix A). Overall, we saw no significant differences in major immune cell populations within the tumors of control and SBS KO cells, suggesting that the phenotypic differences in tumor growth were tumor cell-intrinsic and not due to an altered immune response in the TME.

## 4. Discussion

IFNγ’s effect on reprogramming has been studied in tumor-associated macrophages, whereby IFNγ affects many important metabolic regulators to alter metabolism [18,23]. Some of the known mechanisms by which IFNγ reprograms the metabolic state of immune cells is by the inhibition of the mammalian target of rapamycin complex 1 (mTORC1), 5′-AMP-activated protein kinase (AMPK), and glycogen synthase kinase 3 (GSK3) [24,25]. In this study, we found that IFNγ induces NAMPT in both human and mouse melanoma cells. Further, we identified the importance of IFN/Stat-inducible NAMPT in melanoma growth and proliferation. Moreover, we found that perturbation by IFNγ, in the presence of a NAMPT inhibitor, led to increased apoptosis. We also demonstrated that IFNγ-inducible NAMPT in melanoma cells counteracts the growth-suppressive effects of IFNγ. Notably, Stat1 can sometimes be activated by several other factors besides IFNγ within the TME, which may add valuable insight into the importance of Stat1-inducible NAMPT and will be of interest in future studies. Further, NAMPT is also known to be secreted from cells and is referred to as extracellular NAMPT (eNAMPT), and the mechanism by which IFNγ alters eNAMPT in melanoma will also be of interest for future studies [26,27]. Together, our work describes how IFN/STAT1 can induce NAMPT in human and mouse melanoma cells to increase growth both in vitro and in vivo. It is important to note that some of the mechanisms downstream of IFNγ signaling may be shared between melanoma cells and tumor-associated macrophages [18]. NAMPT orchestrates an effective antitumor immune response in macrophages within the TME, whereas selectively interfering with NAMPT in melanoma cells may improve the efficacy of IFN-based immunotherapies. STING agonists, immune checkpoint blockade (ICB), and other IFN-inducing therapies have been promising for patients with cancer; however, they are not always the frontline or the most effective and durable treatment strategies for melanoma, due to high rates of immune-related adverse events [22,28,29]. These immunotherapy approaches may indirectly upregulate IFN-inducible NAMPT in tumors cells, and we hypothesize that their efficacy can be enhanced by combining these therapies with NAMPT inhibitors that specifically target tumor cells. For instance, increased PD-L1 may lead to immune suppression of antitumor responses; however, increased PD-L1 expression correlates with enhanced immune checkpoint blockade responses, more specifically to anti-PD-1 therapy [30]. In our study, we saw trending increases in PD-L1 expression in non-immune tumor cells without the STAT1 binding site necessary for IFNγ, corroborating a recent study that found that IFN increased PD-L1 expression on liver cancer cells and required IFN-independent NAMPT [31]. Together, these observations implicate increased sensitivity to immune checkpoint blockade therapy and the IFNs induced by immune checkpoint blockade therapy with tumor cell-specific NAMPT inhibition, yet it may depend on the tumor type [31]. Ultimately, our understanding of IFN/Stat-inducible NAMPT may lead to better combinational therapies or improve the efficacy of NAMPT inhibitors. 

## 5. Conclusions

In our work, we were interested in the paradigm between IFNs’ antitumor and protumorigenic roles in melanoma. Altogether, our findings suggest a novel mechanism by which immune molecules alter a key tumor cell metabolic enzyme to enhance tumor survival, which may counteract some of the antitumor effects of IFNs. We believe that targeting protumorigenic IFN-mediated changes in tumor metabolism is critical to the enhancement of IFN-based immunotherapies.

## Figures and Tables

**Figure 1 cancers-15-01411-f001:**
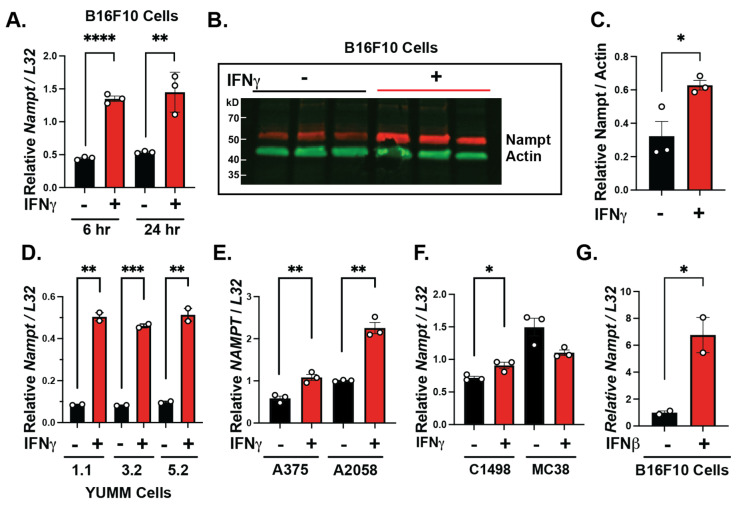
IFNs induced NAMPT in melanoma cells. B16-F10 treated with IFNγ for 6 h and 24 h induced NAMPT RNA gene expression (**A**). NAMPT protein levels were also higher in B16-F10 cells treated with IFNγ using Western blot (**B**) with the expression quantification (**C**). Yale University Mouse Melanoma (YUMM) (**D**) and human melanoma cells all induced NAMPT RNA gene expression when treated with IFNγ for 6 h (**E**). C1498 (leukemia) and MC38 (colon, cancer) cells were treated with 6 h of IFNγ, and NAMPT RNA gene expression was measured (**F**). Type I IFN, IFNβ, led to induced NAMPT RNA gene expression in B16-F10 cells (**G**). n = 3, unpaired *t*-test, mean ± SEM; * *p* ≤ 0.05, ** *p* ≤ 0.01, *** *p* ≤ 0.001, **** *p* ≤ 0.0001. Experiments are representative of at least two independent experiments.

**Figure 2 cancers-15-01411-f002:**
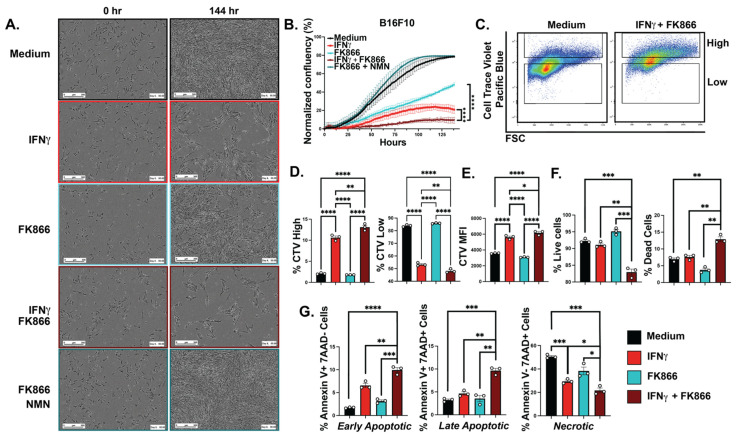
IFNγ and FK866 led to decreased melanoma cell proliferation and increased apoptosis. Representative images at the initial (0 h) and final timepoints (144 h) of B16-F10 cells were treated with medium, IFNγ (50 ng/mL), FK866 (50 µM), IFNγ + FK866, and FK866 + NMN (100 µM) and imaged using Incucyte live imaging analysis; scale bars represent 300 µM (**A**). Percent confluency of B16-F10 cells under various treatment types were graphed over time; representative data from single experiment, center values are mean ± SEM, one-way ANOVA (**B**). Gating strategy for B16-F10 cells with Cell Trace Violet (CTV) showing the CTV high and CTV low populations in medium and IFNγ (50 ng/mL) + FK866 (50 µM)-treated cells (**C**). Percent of B16-F10 cells treated with CTV under various drug conditions (**D**) and mean fluorescence intensity of Cell Trace Violet (**E**); representative data from single experiment, unpaired *t*-test, mean ± SEM. Percent of live and dead B16-F10 cells with medium, IFNγ (50 ng/mL), FK866 (50 µM), and IFNγ + FK866 using ghost dye (**F**) and percent of early and late apoptotic and necrotic B16-F10 cells using Annexin V and 7AAD+ staining; representative data from single experiment (**G**), unpaired *t*-test, mean ± SEM. * *p* ≤ 0.05, ** *p* ≤ 0.01, *** *p* ≤ 0.001, **** *p* ≤ 0.0001. Experiments are representative of at least two independent experiments.

**Figure 3 cancers-15-01411-f003:**
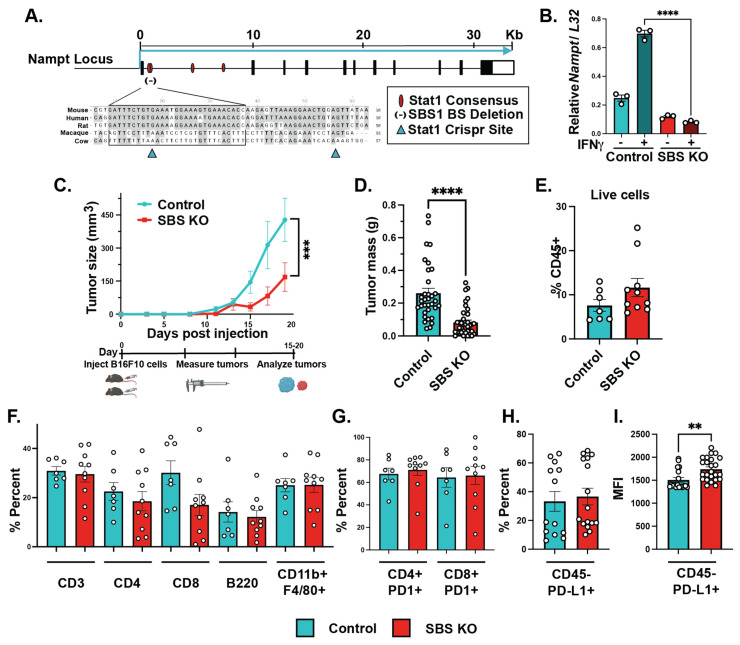
STAT1 binding site KO B16-F10 tumors grew significantly smaller than their control counterparts. Graphical schematic of SBS KO B16-F10 cells created using a gRNA targeting a STAT1 binding site on the NAMPT gene and a sgRNA for the control cells using Crispr Cas9 (**A**). Quantitative-PCR results demonstrating the ratio of NAMPT induction in response to IFNγ using RPL32 to normalize gene expression of control, SBS KO, and native B16-F10 cells; n = 3; center values are mean ± SEM; two-way ANOVA with multiple comparisons (**B**). Graphical schematic of methods where WT B6 mice were injected subcutaneously into the flank with either control or SBS KO B16-F10 cells. Tumor growth was measured every other day for the duration of the experiment, and representative data from multiple experiments are shown; n = 6; center values are mean ± SEM; two-way ANOVA with multiple comparisons (**C**). Final tumor mass, control tumors n = 36 and SBS KO tumors n = 46; data were combined from multiple individual experiments; outliers were excluded using the ROUT method; center values are mean ± SEM; unpaired *t*-test (**D**). We quantified the percent of T cells (CD45+ CD3+), helper T cells (CD45+ CD3+ CD4+), cytotoxic T cells (CD45+ CD3+ CD8+), B cells (CD45+ B220+), macrophages (CD45+ CD11b+ F4/80+), and PD1 expression on helper T cells (CD45+ CD3+ CD4+); representative data from single experiment; n = 6; center values are mean ± SEM; unpaired *t*-test (**E**–**G**). While trending increases were observed, there was no significant change in the percent of PD-L1 expression (**H**), while the MFI had modest increases (**I**) in tumor cells (CD45- PD-L1+); data from two experiments; control n = 13; SBS KO n = 16; center values are mean ± SEM; unpaired *t*-test (**H**,**I**). ** *p* ≤ 0.01, *** *p* ≤ 0.001, **** *p* ≤ 0.0001. Experiments are representative of two independent experiments.

## Data Availability

The data presented in this study are available on request from the corresponding author. The data are not publicly available due to institutional guidelines.

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
