# Peer review of "Interferon Gamma-Inducible NAMPT in Melanoma Cells Serves as a Mechanism of Resistance to Enhance Tumor Growth"

_cancers, 2023, doi:10.3390/cancers15051411_

Round 1

Reviewer 1 Report

This study is a follow-up study from the same group published in Nature Communications, 2021, in which they identified a novel Stat1 binding element in the promoter locus of the nampt gene of macrophages which encodes nicotinamide phosphoribosyltransferase for NAD+ production. Herein, the authors changed the research system from macrophages to melanoma cells and demonstrated that melanoma cells responded robustly to IFNγ by increasing the expression level of NAMPT, as well as improving tumor cell fitness and proliferation in vivo. Several questions need to be addressed before the consideration of possible publication in Cancers.

1.     Figure 1B: The authors have to run assays again to obtain a high-quality gel picture. For the revised experiment, the author should also include MC38 cells as a negative control.

2.     Figure 2F-2G: The authors apparently forgot to describe these data in the manuscript.

3.     Figure 3B: In the absence of IFNγ, there was a ~50% decrease in mRNA expression of nampt gene in SBS KO B16F10 cells compared to the control cells, which raised reasonable speculation to the reviewer that cytokines other than IFNγ can also drive JAK-STAT1-NAMPT transcriptional activation axis. This point is critical for the interpretation of the contradictory results between in vitro assay and in vivo assay. In vitro, IFNγ showed a clearly suppressive effect on tumor cell growth (The entire Figure 2). Although the authors ruled out many parameters from tumor-infiltrated immune cells and concluded it was a cancer cell-intrinsic effect on tumor shrinkage due to the lack of a key SBS binding site (Figure 3C-G), the data only supported the necessity of STAT1-mediated expression of NAMPT in the proliferation of B16F10 cells in vivo. With the current dataset, the authors can not conclude IFNγ within the TME is the major driving force to promote the expression of NAMPT in melanoma cells through STAT1. With this in mind, the conclusion of the whole manuscript needs to be downgraded or changed.

4. Figure 3B: The corresponding Western Blot result should also be included.

Author Response

This study is a follow-up study from the same group published in Nature Communications, 2021, in which they identified a novel Stat1 binding element in the promoter locus of the nampt gene of macrophages which encodes nicotinamide phosphoribosyltransferase for NAD+ production. Herein, the authors changed the research system from macrophages to melanoma cells and demonstrated that melanoma cells responded robustly to IFNγ by increasing the expression level of NAMPT, as well as improving tumor cell fitness and proliferation in vivo. Several questions need to be addressed before the consideration of possible publication in Cancers.

  1. Figure 1B: The authors have to run assays again to obtain a high-quality gel picture. For the revised experiment, the author should also include MC38 cells as a negative control.

Thank you for the comment. We have corrected and included this western blot as a raw scan image to address the quality concerns in figure 1. Additionally, we have included a quantification of the western blot, which shows clear differences in Nampt protein levels between the groups (Figure 1C).

  1. Figure 2F-2G: The authors apparently forgot to describe these data in the manuscript.

Thank you for the observation. We have now added relevant descriptions to the text.

  1. Figure 3B: In the absence of IFNγ, there was a ~50% decrease in mRNA expression of namptgene in SBS KO B16F10 cells compared to the control cells, which raised reasonable speculation to the reviewer that cytokines other than IFNγ can also drive JAK-STAT1-NAMPT transcriptional activation axis. This point is critical for the interpretation of the contradictory results between in vitro assay and in vivo assay. In vitro, IFNγ showed a clearly suppressive effect on tumor cell growth (The entire Figure 2). Although the authors ruled out many parameters from tumor-infiltrated immune cells and concluded it was a cancer cell-intrinsic effect on tumor shrinkage due to the lack of a key SBS binding site (Figure 3C-G), the data only supported the necessity of STAT1-mediated expression of NAMPT in the proliferation of B16F10 cells in vivo. With the current dataset, the authors can not conclude IFNγ within the TME is the major driving force to promote the expression of NAMPT in melanoma cells through STAT1. With this in mind, the conclusion of the whole manuscript needs to be downgraded or changed.

We do not interpret the in vitro vs. in vivo results to be contradictory. In vitro, the suppressive effects of IFNg are enhanced when Nampt is inhibited, consistent with Nampt providing some protumor effects when melanoma cells are exposed to IFNg. In vivo, tumors grow slower through disruption of Stat1 (of which IFNg is a major activator) mediated induction of Nampt. However, we acknowledge that several factors can activate Stat1 in addition to IFNg in the TME, and have expanded our discussion to include such possibilities.

  1. Figure 3B: The corresponding Western Blot result should also be included.

We have consistently found that Nampt RNA changes reflect protein changes, as shown in Figure 1 and in our previous publication (Huffaker et al. Nature Communications 2021). Thus, we expect that the mRNA differences shown in 3B are reflected at the protein level. Further, this is functionally supported by the reduced tumor growth that correlates with lower Nampt mRNA levels.

Reviewer 2 Report

The manuscript by Barba et al. has shown the importance of Nampt expression mediated by IFNg in melanoma cell line growth and proliferation. 

The authors have already published in Nature Communication a comprehensive study in which they highlighted NAMPT as a target of STAT1 during tumour associated macrophages (TAMs) by interferon gamma.

This manuscript tries to connect the findings of the precedent work on TAMs and the tumour cells.

There are some limitations of the review that the authors should address to better describe this phenomenon.

Major revision:

1. In the introduction the authors mentioned the intracellular role of NAMPT only. At the same time, they have shown in Figure 1B the presence of NAMPT in the medium of melanoma cell lines, without explaining why they showed it and monitored it, and therefore significance of the presence of extracellular NAMPT (eNAMPT). Few words about eNAMPT in the introduction and its possible release induced by IFNg from all the cell lines they have used in Figure 1 need to be added. Moreover, other groups have highlighted the importance of IFNg in eNAMPT release. Also in this case, this papers need to be mentioned (Colombo et al.,2022 DOI: 10.1016/j.cmet.2020.10.021)

2. The authors have shown that no difference in PD-1 expression on T cells has been highlighted. What about PD-L1 expression on B16-SBS KO cell line?

3. Also another group (Lv et al, 2021 DOI: 10.1016/j.cmet.2020.10.021) has identified the connection between Stat1 and IFNγ-mediated induction of Nampt in tumor cell line. This work needs to be mentioned and connected with the findings of this work.

Minor revision.

1. In the abstract IFNg hasn't been written with the greek letter.

2. A gating strategy for flow cytometry needs to be added in the Supplementary part.

Author Response

The manuscript by Barba et al. has shown the importance of Nampt expression mediated by IFNg in melanoma cell line growth and proliferation. 

The authors have already published in Nature Communication a comprehensive study in which they highlighted NAMPT as a target of STAT1 during tumour associated macrophages (TAMs) by interferon gamma.

This manuscript tries to connect the findings of the precedent work on TAMs and the tumour cells. 

There are some limitations of the review that the authors should address to better describe this phenomenon.

Major revision:

  1. In the introduction the authors mentioned the intracellular role of NAMPT only. At the same time, they have shown in Figure 1B the presence of NAMPT in the medium of melanoma cell lines, without explaining why they showed it and monitored it, and therefore significance of the presence of extracellular NAMPT (eNAMPT). Few words about eNAMPT in the introduction and its possible release induced by IFNg from all the cell lines they have used in Figure 1 need to be added. Moreover, other groups have highlighted the importance of IFNg in eNAMPT release. Also in this case, this papers need to be mentioned (Colombo et al.,2022 DOI: 10.1016/j.cmet.2020.10.021)

To clarify the confusion with our labels, we changed the labels for Figure 1B. The western blot shown is taken from whole cell lysate where we removed the supernatant, so we are assaying cellular Nampt and not eNAMPT. However, extracellular NAMPT is something we have thought about and we appreciate this feed-back; therefore, we have included this in the discussion.

The authors have shown that no difference in PD-1 expression on T cells has been highlighted. What about PD-L1 expression on B16-SBS KO cell line?

This is a good point and something we thought about. We injected control and SBS KO melanoma cells into mice and quantified PD-L1 positive CD45 negative cells after harvesting the tumors. We saw trending increases in the percent of PD-L1 positive cells and modest but significant increases in the mean fluorescent intensity (MFI) of the cells. (control tumors n=13, SBS KO tumors n=16). We have added these data to the manuscript (Fig 3H-I).

  1. Also another group (Lv et al, 2021 DOI: 10.1016/j.cmet.2020.10.021) has identified the connection between Stat1 and IFNγ-mediated induction of Nampt in tumor cell line. This work needs to be mentioned and connected with the findings of this work.

Consistent with our work mentioned above, the authors in Lv et al. describe that liver cancer cells that have inhibited Nampt grow significantly smaller tumors than the control counterpart. Furthermore, the authors in that paper describe an indirect way by which perturbation of NAD+ metabolism, specifically Nampt inhibition, leads to less Stat1 phosphorylation and activation. We show a direct relationship between IFNg signaling leading to STAT1 binding on the Nampt gene to increase Nampt production. Both results are important to consider as there could be multiple pathways by which IFNg signaling and NAD+ metabolism are connected. We have added more regarding this in the discussion and believe these manuscripts work together to further emphasize the importance of Nampt in tumor biology.  

Minor revision.

  1. In the abstract IFNg hasn't been written with the greek letter.

Thank you for that detail, it has been corrected.

  1. A gating strategy for flow cytometry needs to be added in the Supplementary part.

Thank you, we have added this gating strategy to the supplemental section (Supplemental Figure 2).

Reviewer 3 Report

In this second version of the manuscript by Barba & Co-workers, I still found some incorrections in the text description aligned with the figures, quality of images, plots, and conclusions taken by the authors. 

Here I specify my revision points:  

1-The quality of the graphical abstract should be improved; It's blurred and distorted, especially the text but also the images, maybe as a result resolution of the image used for the PDF version.

2- Line 103 should mention the composition of the media used to culture both B16 and human melanoma cell lines.  As an alternative, use a reference to a previous paper where those were described. 

3- Line 39 should be addressed the number of animals used in each condition.

4- Line 170 there is a mistake in the sentence -"were cultured t a seeding density". Should be something like were cultured at a seeding density of...

5 -Line 184 there is a misspelled word; Additionally, melaomas

6- Line 186 the authors claim that  IFNγ treatment for 24 hours induced higher levels of the Nampt protein in B16F10 cells (Fig 1B). The fact that authors separated the two distinct time point results into distinct plots and the Y-axis is different between them, creates the illusion that after 24h treatment of IFNg induces higher levels of Nampt. In reality, the expression levels of Nampt in the two distinct time points are not significantly different. I advise the authors to revise the text or alter the plot presentation to be more clear and more objective.

7- In the legend of figure 1 and the text this version of the manuscript contains mistakes; i.e., figure 1D is not the expression plot of A375 cells but YUMM; figure 1F is the one where the expression was measured in other types of cancers (colon and leukemia) and in text, it is referenced as figure 1E. Also this figure description, starts with the nomenclature of (Fig. 1D) and then changes to (Fig. 1d). More attention should be given to the correction of these details.  

8- Line 201 should be written -After observing IFN induction of Nampt expression in melanoma cells and in line 202 the treatment was with FK866 and not IFNg. It is written," with IFNγ, FK866,".

9- The quality of the images in figure 2 is very poor as also the plots and graphs. Again could be a not sufficient resolution but it is not distinguishable colors, letters, or cells in the images. Must be revised and changed since this present format is not sufficient for publication in this journal.

10- In conclusion, in line 290, I disagree with the author's statement that "our findings provide a novel mechanism by which immune molecules alter a key tumor cell metabolic enzyme to enhance tumor progression through counteracting some of the antitumor effects of IFN". 

Based on the presented data the authors can only conclude that this work shows how IFN/Stat1γ can induce Nampt in human and mouse melanoma cells to increase growth in vitro and in vivo.  The results of figure 3 show that no significant differences were obtained in immune cell populations within the tumors of control and SBS KO cells, showing that the differences in tumor growth were tumor cell-intrinsic and not due to an altered immune response in the TME. 

In summary, I ask the authors to revise the points that were described before the manuscript can be considered for publication.

Author Response

Word document is attached. 

Round 2

Reviewer 1 Report

As the authors have not completed my request for revisions to this manuscript, I cannot be more positive for the publication of this work in Cancers.

Author Response

We have now thoroughly considered all the comments and done our best to address them. 

Reviewer 1 requested a high quality image of the western blot shown in figure 1. To address this, we included a raw scan and a clear infrared image with western blot quantification to validate the significant increase in Nampt expression in the presence of IFNg in melanoma cells. Another concern that we addressed was validation of SBS KO cells having less Nampt in response to IFNg. For this, we clearly show that Nampt mRNA expression is induced by IFNg in Wt Melanoma cells, yet substantially diminished in IFNg treated SBS KO cells. This analysis was done prior to using these cell lines for our in vivo tumor growth experiments to ensure that the SBS KO cells had a diminished Nampt response to IFNg that is produced by immune cells in the TME. Further, we also point to data in our previous publication (Huffaker et al Nat. Commun. 2021) which, in addition to our data in Figure 1, demonstrates that Nampt mRNA correlates with protein levels which is why we did not also perform Western blots using these cells. Lastly, the reviewer commented on a few points in the discussion and, in response, we now mention the possibility that other activators of Stat1 might also impact Nampt expression by tumor cells. However, we feel that our conclusion that Nampt is induced by IFNg/Stat1 signaling in melanoma cells, and its induction impacts tumor growth and survival is strongly supported by our data.

All together, we appreciate these comments and insights, and believe that addressing them has improved the manuscript.

Reviewer 2 Report

Accept in present form

Author Response

Thank you

Reviewer 3 Report

I believe the authors address in time the queries made in the previous revision of the manuscript and It is my opinion to accept it for publication in this present version but with a final revision of gaps and spelling mistakes.